# OpenReview forum: "Beyond External Monitors: Enhancing Transparency of Large Language Models for Easier Monitoring"
_ICML.cc/2026/Conference — ICML 2026 regular_

### Official Review · Reviewer_SSVr · 2026-03-07

**Soundness:** 2
**Presentation:** 2
**Significance:** 4
**Originality:** 3
**Overall Recommendation:** 3
**Confidence:** 3

**Summary:**

This submission proposes a method called TELLME that fine-tunes LLMs to have internal representations that cluster together for the same behavior and are disentangled for different behaviors. Given data samples from multiple types of behavior, TELLME optimizes existing disentanglement losses to achieve disentanglement, while also optimizing a loss on a “retain” dataset to preserve general capabilities. Models trained using TELLME are evaluated in terms of the ability to monitor them for safety risks, their toxicity, and their general performance. The experiments span LLMs and vision-language models (VLMs), dense and mixture-of-expert (MoE) models, and a range of model sizes. The results show improved monitoring performance compared to linear probe and LLM-as-a-judge baselines, improved toxicity compared to supervised fine-tuning (SFT), and also evidence for clustering of representations.

**Compliance With Llm Reviewing Policy:**

Affirmed.

**Final Justification:**

I am not comfortable enough to move upward from my initial assessment of Weak Reject. In particular, I am not quite satisfied with the responses to my concerns 1 and 2 under "Soundness":
- Section 6 theoretical analysis: I think that this section needs to be rewritten to tell a coherent story: 1) narrow the scope to monitoring of LLM behavior (which is more clearly modelled as a classification problem), 2) clearly state the generalization bound of Chuang et al. (2021) that applies to this setting, 3) show the evidence that TELLME reduces a quantity in this bound, and 4) show that TELLME improves other measures of intra-class compression and inter-class separation. The claims in the abstract and introduction about "theoretical analysis" also need to be revised.
- Figure 3: While I now understand that the MMLU-Pro and MMMU scores reported in the rebuttal were obtained through a different evaluation protocol, the large difference between these values and the ones plotted in Figure 3 still gives me pause.

In addition, I feel that a review of a revised manuscript is needed to fully address my concern 1 under "Presentation" (including the sub-points under it). This is true to a lesser extent for my concerns 3 and 4 under "Soundness" as well.

Overall, I was less satisfied with the authors' first rebuttal to me. While I was more satisfied with their second rebuttal, the first rebuttal decreased my confidence that the revisions could be carried out without requiring another round of review.

**Key Questions For Authors:**

Please see the comments under Presentation, especially comment 3 on the experimental setup.

**Limitations:**

Yes

**Strengths And Weaknesses:**

### Strengths
- The specific idea of training LLMs (and VLMs) to have disentangled representations appears to be new to my knowledge (although disentangling representations and training more interpretable LLMs are not new by themselves)
- The scope of the experiments is impressive: LLMs and VLMs, several model families and a large range of sizes for Qwen models, multiple types of safety risks, multiple benchmark datasets, etc.
- The improved toxicity/safety that appears to be a natural consequence of TELLME, without further measures, is particularly noteworthy

### Weaknesses
**Soundness**: I think some claims are overstated. Having said that, I do not think that the degree of overstatement is that serious, or it affects more secondary claims, so that it should not be difficult to remedy.
1. I think that the claim of “theoretical analysis” in Section 6 is overstated. I see only an argument that TELLME *should* decrease two quantities ($k$-variance and $\tau$-margin loss) present in the existing generalization bound of Chuang et al. (2021). But there is not even empirical evidence that it does decrease these quantities. It is also not clear what is meant by “generalization ability” here.
1. Figure 3: While I think it is fine to use previously reported MMLU-Pro scores and MMMU scores for the base models, the MMLU-Pro and MMMU scores for the corresponding TELLME models should be actually computed. In that way, one would be able to see the effect of TELLME on both axes, safety and general capability, similar to Table 2. Currently, it looks like the TELLME points are simply aligned horizontally with the corresponding base model points.
1. I think the “10.2% better” claim in line 250 should be tempered. First, it is not clear what 10.2% refers to (between which methods). Moreover, I think the most meaningful comparison is among the linear probe methods because they are generally the strongest, much stronger than Self-Sim. But the improvement of TELLME for linear probes is more modest than 10%.
    1. A more minor related comment: What is the evidence for the 1.1% improvement in LLM-as-a-judge accuracy being due to better comprehension?
    2. Another similar comment: It is not clear what “8.1% better safety” in lines 311-312 refers to specifically.
1. I find the evidence for the claim of transparency a bit limited, being only the t-SNE visualizations in Figure 4 in the main paper, although there is a reference to quantitative clustering metrics in Appendix A.2. Perhaps the claim of transparency should be weakened to one of “clustering” or “disentanglement” of representations, and/or transparency as needed for safety.

**Presentation**:
1. There are many aspects of the description of TELLME that are unclear just based on Section 3.2. Algorithm 1 in the appendix clarifies most but not all of these aspects. It would be good to not only move Algorithm 1 into the main paper, but also add more explanatory text and improve notation.
    1. It is not clear that $B$ is the batch size in each iteration.
    2. It would be good to add a brief explanation of the InfoNCE loss in (3) to make the paper more self-contained.
    3. The subscript $\mathbf{x}_{c_k}^{i_1}, \mathbf{x}_{c_k}^{i_2}$ in (3) is confusing. Algorithm 1 seems to state that one pair of $\mathbf{x}$’s is sampled from each of the $B$ behaviors in each iteration. If this is true, then I think it would be clearer to rewrite (3) as just the loss for one batch. The expectation is then implied.
    4. It is not clear that a single layer $l$ and a single token position $t$ are selected, i.e., these are hyperparameters and there is no sum over $l$ for example. It is also not clear why the layer at 80% depth is chosen as the target, although hints are given later in the paper.
    5. The subscript ${\mathbf{x}^k}_{k=1}^B$ in (4) is similarly confusing. At a minimum, the notation should indicate that $\mathbf{x}^k$ is drawn from the Retain Set while the $\mathbf{x}^i$’s are drawn from the Disentangle Set.
2. The justification for the KL penalty should be made clearer earlier. Currently, the paragraph in lines 194-201 does not help. The Training Setup paragraph in Section 4.2 partly clarifies things, in that the KL penalty should be removed if we want the TELLME model to respond differently (i.e., more safely) than the base model.
3. Some aspects of the experimental setup are also unclear:
    1. How is TELLME performed on the last token of both input and output sequences? How are the two tokens combined?
    2. How is LoRA used? Is the TELLME model the base model plus the LoRA adapter, and only the LoRA parameters are fine-tuned to optimize the TELLME objective?
    3. Are the same datasets (BeaverTails, etc.) and behaviors used for training TELLME as for evaluation? The Training Setup paragraph in Section 4.2 suggests so.
    4. The sentence beginning with “Without telling LLMs …” in lines 269-273 is unclear, as is the phrase “for the potential improvement expected in Section 4.1” before it. If it is indeed the case that TELLME training is applied to the LLM and no further measures for safety, then this first paragraph of Section 4.2 should say so more explicitly.

Minor:
- I think the mentions of “scalable oversight of artificial super intelligence” are purely speculative and should be removed. It is also not clear what some uses of the word “scalability” refer to, specifically “scalability of TELLME” in lines 101-102 and “impressive scalability” in lines 437-438.
- Regarding the definition of LLM transparency in Section 3.1, 1) I do not see why “data sources” are a necessary part of the definition, and 2) LLM thinking may not “correspond closely to human understanding” but can still be transparent, i.e., these are separate properties.
- Section 5, first paragraph: It would be better to at least mention which hyperparameters are investigated in the appendix.
- Table 3: I think it would be clearer to put “Original” and “TELLME” in the leftmost column (the one currently labelled “Setting”) and reserve the “Components” column for components of TELLME’s retain loss.
- Lines 426-427, “two models”: I see four models in Figure 4

---

> ### Author Rebuttal · Authors · 2026-03-31
>
> Thank you for your great efforts in reviewing this paper. We will try our best to answer all questions. **Please let us know if you still have further concerns, or if you are not satisfied with the current responses, so that we can further update the response ASAP.**
>
> ---
> **Q1**: Comment 1 under soundness for the theoretical analysis.
>
> A1: Thank you. Our analysis approximates LLM generation as a behavior-level classification task (Lines 367–370), providing a theoretical motivation: intra-class compression (↓k-variance) and inter-class separation (↓τ-margin loss) should be improved, which guided us to measure five proxy disentanglement metrics (Lines 425–438). Nevertheless, we **have followed your suggestions to conduct new experiments** measuring an empirical proxy of k-variance (mean intra-class representation variance): for Llama-3.1-8B on the binary safety task, it decreases from 7.3e-4 (Origin) to 2.5e-4 (TELLME), supporting the theoretical analysis.  We will revise Section 6 to make the approximation explicit.
>
> ---
>
> **Q2**: Comment 2 under soundness for “real MMMU score”.
>
> A2: Thank you. The purpose of Figure 3 is to show that TELLME remains effective on models with different capability levels, rather than to use that figure itself as evidence that TELLME preserves general capability.
>
> **We have Followed your suggestion to additionally evaluate** the corresponding TELLME models on the capability axis. Due to computational constraints, we are unable to evaluate the 235B model on MMLU-Pro; however, the Qwen2.5-72B result (MMLU: 71.8→71.7) already demonstrates that TELLME maintains general capability at large scale. As safety improvements are shown in Figure 3 and Table 9, we here report capability scores:
>
> ||MMMU||MMLU-PRO||
> |-|-|-|-|-|
> ||Origin|TELLME|Origin|TELLME|
> |Qwen2.5|||||
> |7B|||57.1|57.1|
> |14B|||65.0|65.1|
> |32B|||69.6|69.1|
> |72B|||71.8|71.7|
> |Qwen3-VL|  ||||
> |8B|56.1|55.4|64.5|64.6|
> |30B|65.3|64.3|70.7|69.6|
> |32B|67.3|68.6|74.4|74.2|
>
> ---
> **Q3**: Comment 3 under soundness for the reported improvement.
>
> A3: Thank you. "10.2% better" is the average gain over Origin across Self-Sim and LP; for LP alone, +5.6% over Origin and +2.5% over the strongest baseline. "8.1% better safety" is the absolute average improvement over Origin on BT+SB. The 1.1% Judge gain is included not to claim a large accuracy improvement, but to show that representation-level disentanglement already begins to influence LLM output behavior -- serving as motivation for the detoxification in Section 4.2. We will make all baselines and breakdowns explicit in the revision.
>
> ---
> **Q4**: Comment 4 under soundness for the claim of transparency.
>
> A4: Thank you for the suggestion. In our paper, clustering/disentanglement is the mechanism by which TELLME improves transparency for "monitoring"(Lines 147–164). Beyond the t-SNE in Figure 4, Appendix A.2 (Table 7) reports five quantitative metrics that all improve consistently across three model families: Coding Rate (↓6–2%), eRank (↓72–83%), L2 distance (↑3.5×–6.7×), inter-class Angle (↑2.5×–3.2×), and Hausdorff distance (↑4.0×–4.1×). These provide strong quantitative support for the disentanglement claim. We will clarify this connection in the revision.
>
> ---
> **Q5**: Comment 1 and 2 under Presentation about notations.
>
> A5: Thank you. We will rewrite Eq.(3)-(4) as a batch loss, clarifying that B is the batch size. Notation: superscript-only ($x^k$) = Retain Set; behavior-subscript ($x_{c_k}^{i_1}$) = Disentangle Set. Layer choice (80% depth) and last-token selection both follow standard representation engineering practices [s1-2] as stated in Lines 1026-1028. Strict page limits forced Algorithm 1 into the appendix — the root cause of the issue you identified. We may move Related Work to the appendix to free space for Algorithm 1 in the main text. We agree the KL penalty motivation is not explained early enough and will move this explanation earlier in the revision.
>
> [s1] Improving alignment and robustness with circuit breakers
> [s2] Inference-time intervention: Eliciting truthful answers from a language model
>
> ---
> **Q6**: Comment 3 under Presentation
>
> A6: Thank you. To clarify the experimental setup: (1) TELLME losses are computed separately on the last tokens of the input and output sequences and then averaged. (2) The base model remains frozen; only the added LoRA adapter parameters are trained. (3) For Section 4.2, we train TELLME exclusively on the BeaverTails training split and evaluate on its test split, alongside SaladBench for a comprehensive safety assessment. (4) As you noted, Lines 269-273 indicate TELLME is applied directly with no additional safety-specific measures. We will explicitly incorporate all these implementation details into the revised manuscript.
>
> ---
> **Q7**: Minor Issues
>
> A7: Thank you. For “scalability,” please see our A1 to Reviewer 2nJK. For the remaining wording, definition, and table/figure issues, we will follow your suggestions in the revision.

---

> > ### Author Rebuttal · Reviewer_SSVr · 2026-04-03
> >
> > Thank you for the rebuttal. However, several issues remain for me.
> >
> > **Q1:** I do not think that Section 6 should be characterized as "theoretical analysis." First, it is not clear how LLM generation is approximated as behavior-level classification. To me, the classification problem is to classify the LLM's behavior, which is a completely separate problem from how the LLM generates tokens. Second, assuming that "generalization ability" refers to the generalization of this behavior classifier, Section 6 simply cites the theoretical result of Chuang et al. (2021). While the rebuttal now provides evidence in a limited case that one quantity (an empirical proxy of $k$-variance) in Chuang et al.'s bound decreases, this is not theoretical analysis to me. In my view, Section 6 is more about understanding how/why TELLME enables better behavior classification.
> >
> > **Q2:** The new MMLU-Pro and MMMU scores reported in the rebuttal do not align at all with the values plotted in Figure 3. Moreover, if "the purpose of Figure 3 is [only] to show that TELLME remains effective on models with different capability levels," then I recommend that the figure be presented in a different way, not by plotting MMLU-Pro and MMMU scores on the x-axis unless those scores can be correctly computed.
> >
> > **Q3:** Indeed, all of the percentage improvement claims should be made more specific. Regarding the 1.1% judge gain, my comment was not about its size but rather about the claim that it is due to enhanced model comprehension.
> >
> > **Q4:** I understand, but I think that "transparency" should be qualified with "for monitoring" or similar, or sometimes substituted with "clustering/disentanglement."
> >
> > **Q5:** Nothing more to say since I cannot evaluate the revisions of course.
> >
> > **Q6:** These questions were addressed.

---

> > > ### Author Response · Authors · 2026-04-04
> > >
> > > Thank you for the follow-up comments and for engaging deeply with our work! We sincerely appreciate the opportunity to clarify and accordingly improve this work.
> > >
> > > ---
> > >
> > > **Q1**: "I do not think that Section 6 should be characterized as "theoretical analysis."..."
> > >
> > > **A1:** Thank you for the clarification — we now better understand your concern. We agree that "theoretical analysis" is not the right characterization for Section 6, as it does not present new theoretical results. As you suggest, Section 6 is better understood as an analysis of how and why TELLME enables better behavior classification, drawing on the existing generalization bound from Chuang et al. (2021).
> > >
> > > To explain our original reasoning behind the classification approximation: we observed that several downstream tasks can be viewed through a classification lens — for example, in detoxification, the LLM essentially performs prompt-level binary classification (safe->respond vs. unsafe->refuse), and monitoring is an even more natural classification problem where a monitor categorizes the LLM's behavior based on its representations.
> > >
> > > However, we agree with your point that this approximation is a stretch when applied to general LLM generation. In the revision, we will narrow the scope of Section 6 to the monitoring task, which is an explicit classification problem, and reframe it as an analysis of how/why TELLME enables better behavior classification rather than "theoretical analysis."
> > >
> > > ---
> > >
> > > **Q2**: "The new MMLU-Pro and MMMU scores reported in the rebuttal do not align at all with the values plotted in Figure 3..."
> > >
> > > **A2:** Thank you for catching this. The remaining discrepancy with Figure 3 is due to different evaluation sources: as stated in Lines 352-359 of the manuscript, Figure 3's x-axis uses MMLU-Pro and MMMU scores *quoted from the models' official technical reports*, since we only needed a capability-level reference for each model. In the rebuttal, however, we needed to evaluate both the Origin and TELLME-finetuned models under identical conditions, so we *used the OpenCompass codebase to obtain comparable before/after evaluation scores*. Different evaluation protocols (prompting format, few-shot setting, etc.) naturally lead to different absolute values from the technical reports.
> > >
> > > Additionally, we have now completed the evaluation of Qwen3-VL-235B, which was previously unavailable due to computational constraints:
> > >
> > > |          | MMLU-Pro | MMMU |
> > > | -------- | -------: | ---: |
> > > | 235B     |     77.6 | 68.7 |
> > > | + TELLME |     77.1 | 68.0 |
> > >
> > > The 235B result shows only a marginal capability drop at the largest scale we tested.
> > >
> > > We agree with your suggestion that Figure 3 should not plot MMLU-Pro/MMMU on the x-axis. In the revision, we will change the x-axis to model parameter count, which unambiguously reflects model scale. We will report the OpenCompass MMLU-Pro and MMMU evaluation results in a separate table.
> > >
> > > ---
> > >
> > > **Q3**: "Indeed, all of the percentage improvement claims should be made more specific. Regarding the 1.1% judge gain, my comment was not about its size but rather about the claim that it is due to enhanced model comprehension."
> > >
> > > **A3:** Thank you for the clarification. We agree that attributing the Judge gain to "enhanced model comprehension" is not well-supported. We will remove the "comprehension" attribution in the revision and instead state that the small Judge gain suggests disentanglement may influence generation behavior, which motivates the more targeted detoxification in Section 4.2.
> > >
> > > ---
> > >
> > > **Q4**: "I understand, but I think that "transparency" should be qualified with "for monitoring" or similar, or sometimes substituted with "clustering/disentanglement.""
> > >
> > > **A4:** Thank you. In the revision, we will consistently qualify "transparency" as "transparency for monitoring" and use "disentanglement" when referring specifically to the representation-space property.

---

### Official Review · Reviewer_n1Y5 · 2026-03-09

**Soundness:** 4
**Presentation:** 4
**Significance:** 4
**Originality:** 4
**Overall Recommendation:** 6
**Confidence:** 3

**Summary:**

This paper introduces TELLME, a method for improving the monitorability of LLMs’ internal representations by increasing the separation between representations of desirable and undesirable `thoughts’. Prior work has shown that monitoring the intermediate computations of LLMs to reduce toxic and undesirable output is difficult because LLMs’ internal representations of toxic concepts are entangled with their representations of neutral outputs. This paper proposes a fine-tuning method that increases the separation between toxic and neutral concepts, increasing the performance of existing LLM monitoring techniques. This method is validated on small language models from three different families (Llama, Mistral, and Qwen), where LLMs aligned with TELLME show improvement on recently published alignment and safety benchmarks.

**Compliance With Llm Reviewing Policy:**

Affirmed.

**Final Justification:**

The authors have committed to fixing the presentation issues and my question about Gemma2-9B appearing in some results but not others, so I have raised my score on presentation.

Additionally, the authors conducted some new experiments in their response to reviewer hiTL (multi-layer experiments and providing an example of how TELLME operates on harmful behaviors, instead of words associated with harmful behaviors). In their response to reviewer SSVr, the authors also conduct experiments measuring model capability after TELLME fine-tuning, showing no drop or an insignificant one on the MMLU and MMLU-Pro benchmarks. These improve the paper's soundness.

In light of these new results, I have updated my overall score. I think this is a creative and interesting approach to make LLMs more amenable to monitoring. Additionally, the result that fine-tuning with contrastive loss can be used to disentangle LLMs' internal representations is useful on its own merits.

**Key Questions For Authors:**

1. In Figure 4 (which is an excellent visualization), the model Gemma2-9B is mentioned, but it is not mentioned in Tables 1 or 2. Although it may be impractical (due to compute and space constraints) to add results for Gemma2-9B, can you explain why this model was selected for Figure 4 but not used in other experiments? Explaining this would improve my evaluation of the paper.

**Limitations:**

yes

**Strengths And Weaknesses:**

# Soundness
This paper claims first that TELLME improves the monitorability of LLMs by separating toxic and neutral concepts in representation space, and second that TELLME improves LLMs’ internal capability to avoid toxic outputs.

The first claim is substantiated by the experiments in Table 1, showing that internal monitoring techniques (Self-Sim and Linear Probe) gain higher accuracy on LLMs aligned with TELLME. The difference is especially significant for Self-Sim. The paper compares their approach against recent representation-based approaches using sparse autoencoders and transcoders, which is a fair comparison.

The proposed method uses a contrastive InfoNCE loss to separate an LLM’s internal representations of different safe and toxic behaviors, with a regularization term to keep the fine-tuned LLM from drifting too far from the original model. This is a reasonable approach.

The paper empirically validates their method on a wide range of LLMs from different families, improving the reader’s confidence that TELLME is broadly applicable. The datasets used are recent, peer-reviewed benchmarks appropriate for the task at hand.
The authors also evaluate the fine-tuned LLMs on general-purpose tasks including mathematics and question-answering, showing that TELLME does not significantly degrade LLMs’ capabilities. This experiment is performed over a range of Qwen model sizes, establishing confidence in its applicability to different models.

The paper’s second claim is evaluated in Table 1 using LLM judges to detect toxic vs nontoxic outputs. This shows that although representation-based monitoring is not performed, LLMs aligned with TELLME produce less toxic content.

A **weakness** of this paper: LLM judges are used to evaluate toxic output, but the accuracy of LLM judges on this task is not discussed. Adding a brief discussion of this would improve the paper. Additionally, Figure 4 mentions the Gemma2-9B model but other results, such as those in Tables 1 and 2, do not.

# Presentation
The literature review surveys previous representation-based approaches to monitoring LLMs for toxic outputs. Other approaches discussed include steering vectors and machine unlearning. This section could be improved by adding 1-2 sentences discussing other methods of screening for LLM toxicity such as LLM judges. The authors also make the connection between contrastive learning in computer vision and the proposed approach and mention a recent paper applying contrastive learning to LLMs.

The literature review should be expanded as there are currently a lot of citations without text explanations under “Representations of LLMs”. Adding context for these citations would improve the paper’s readability.
Finally, there is an influential sparse autoencoders paper “Sparse Autoencoders Find Highly Interpretable Features in Language Models” that is not currently cited but may be interesting to the authors.

The paper is clearly written with consistent mathematical notation.

**A formatting issue:** a lot of extra information is provided in the Appendix. In some places, this makes it difficult to read the main paper. Here’s an example from section 4: “For more general tasks, we showcase the improvement of TELLME on the SIQA and the CoLA task from the GLUE benchmark in Tables 12 and 11 of Appendix A.8.”. This could be improved by ensuring references to the Appendix include a short sentence explaining the result or content of the appendix section and defining any acronyms.

# Significance
A strength of this paper is the extensive empirical evaluation on a variety of LLM sizes and families, which produces strong confidence in the results. Experiments on a wide range of small and large LLMs and extensive ablation studies are presented. LLM alignment and monitoring are important areas of research and this paper could be a significant improvement for representation-based monitoring approaches.

# Originality
This paper presents a novel algorithm applying contrastive learning to improve LLM monitorability. It builds on prior literature to develop a theoretically reasonable and empirically well-supported approach.

# Minor Issues
1. The running title at the top of each page is currently “Submission and Formatting Instructions for ICML 2026”
2. In Section 2 there is a typo “of another untrusted LLMs” (should be “of another untrusted LLM”).
3. In Section 2 there is a typo “CoT from may provide unfaithful explanations”
4. In Section 2 there is a typo “Represetation-based monitors”
5. In Section 2 the distinction between “representations of LLMs instead of their output” is not clear.
6. In Section 4.2 and Table 2 the phrase “over-safety” is used but not defined.
7. Table 2 is difficult to parse because of the abbreviations. A stylistic improvement could be to define the abbreviations (“BT”, “ST”, “XST” and which datasets go into the “Average” column) in the table caption, instead of in the “Evaluation Setups”, or to mention in the table caption that the information necessary to understand it is in the “Evaluation Setups”.
8. In section 4.1 the phrase “remains high safety performance with deeper layers” might be a typo for “retains”.
9. In section 4.2 the phrase “TELLME have the potential” should use “has”
10. In section 4.2 the phrase “illuminates the potential of the TELLME” should be “illuminates the potential of TELLME”
11. In section 5 line 378 the phrase “details provided in the Appendix A.7” has an extra “the”
12. In section 5 the phrase “TELLME with whole components of $L_r$” is somewhat confusing. I think “TELLME with the retain loss”  would be clearer.
13. In section 5 the phrase “Please see Appendix A.5, A.6 and A.7 for more details.” should use the plural “Appendices”.
14. Section 6 line 426 says “Figure 4 shows the t-SNE visualization results of two models” but there are 4 models in Figure 4.
15. In the Impact Statement, the sentence “This work aims to advance the field of AI Control and AI Monitor…” should probably use “Monitoring”.
16. In Tables 1, 2, 3, etc. the heading “Origin” should be defined (presumably this is the original model)?

---

> ### Author Rebuttal · Authors · 2026-03-31
>
> Thank you for your great efforts in reviewing this paper. We will try our best to answer all your questions. **Please let us know if you still have further concerns, or if you are not satisfied with the current responses, so that we can further update the response ASAP.**
>
> ---
>
> **Q1**: All minor issues.
>
> A1: Thank you for the careful reading and for pointing out these issues. We appreciate these helpful comments and will correct them in the revised version.
>
> ---
> **Q2**: In Figure 4 (which is an excellent visualization), the model Gemma2-9B-IT is mentioned, but it is not mentioned in Tables 1 or 2. Although it may be impractical (due to compute and space constraints) to add results for Gemma2-9B, can you explain why this model was selected for Figure 4 but not used in other experiments? Explaining this would improve my evaluation of the paper.
>
> A2: Thank you for pointing this out. We originally included Gemma-2-9B because it serves as a critical fairness check against SAE-style baselines. Specifically, as detailed in Appendix A.3 (Lines 928–953 and Table 8), we compared TELLME against Google's open-source SAE on Gemma2-9B. Even though Google's SAE was trained on 8 billion tokens—nearly 1,000 times our training volume—TELLME still achieved a significant advantage. This comparison fully and fairly demonstrates the effectiveness of TELLME.
>
> Furthermore, **we have followed your suggestions** to include new detoxification experiments using Gemma-2-9B. As shown below, the results further validate TELLME’s strong and consistent performance.
>
> | Method                      | BT $\uparrow$ | SB $\uparrow$ | XST $\downarrow$ | Capability $\uparrow$ |
> | --------------------------- | ------------: | ------------: | ---------------: | --------------------: |
> | Origin                      |          98.0 |          97.6 |             20.4 |                  66.8 |
> | + SFT                       |          97.6 |          97.3 |             18.0 |                  64.3 |
> | + TELLME                    |          99.1 |          98.1 |         **14.0** |              **66.7** |
> | + TELLME$_{\text{NT-Xent}}$ |      **99.4** |      **99.0** |             18.4 |                  66.6 |
>
> We acknowledge that featuring Gemma-2-9B in Figure 4 without referencing this Appendix discussion in the main text is confusing. To resolve this and improve the flow of the paper, we will move the Gemma-2-9B  result in Figure 4 to the Appendix in the revised manuscript.
>
> ---
>
> **Q3**: "LLM judges are used to evaluate toxic output, but the accuracy of LLM judges on this task is not discussed."
>
> A3:  Thank you for this helpful comment. As stated in Lines 967-968, we use Qwen3Guard-8B-Gen to evaluate the safety of QA pairs. According to its technical report, its BeaverTails F1 score is 86.6 under the strict setting (and 85.5 under the loose setting), which is among the strongest specialized safety evaluators currently available.
>
> Its classification score may appear lower than the safety rate of the models in our table (even greater than 99), but these quantities are not directly comparable. The reported BeaverTails result is measured on specially constructed ambiguous responses under the same prompt template, whereas real model generations are often much easier to classify in practice (e.g., many unsafe prompts are directly refused).
>
> Moreover, even if the judge's F1 is imperfect, its classification errors are expected to be distributed uniformly across methods, rather than systematically favoring TELLME. The relative ordering of methods therefore remains reliable regardless of the judge's absolute accuracy. We will add this discussion in the revised manuscript to clarify why Qwen3Guard-8B-Gen is an appropriate judge in our setting.

---

> > ### Author Rebuttal · Reviewer_n1Y5 · 2026-04-01
> >
> > **Gemma2-9B:**
> >
> > Thank you for explaining this point. Moving results for Gemma2-9B-IT to the Appendix is a good resolution for the clarity of the paper. Moving the discussion of the SAE baselines in Appendix A.3 to the main paper could also work.
> > In the results table you posted, it seems that TELLME decreases XST but does not noticeably improve on other metrics. So if this table is included in the paper, the results should be interpreted to explain why this happened.
> >
> > **LLM judges:** This is a reasonable justification. Adding it to the paper will improve the clarity of the experimental design.
> >
> >
> > To follow up on an issue from the original review: a lot of extra information is provided in the Appendix. This wasn’t mentioned in the authors’ rebuttal but it was also not originally placed in the key questions of the review. However, it remains a concern for the readability of the paper.

---

> > > ### Author Response · Authors · 2026-04-02
> > >
> > > Thank you for the follow-up comments and for engaging deeply with our work! We sincerely appreciate the opportunity to clarify and accordingly improve this work.
> > >
> > > ---
> > >
> > > **Q:** "A formatting issue: a lot of extra information is provided in the Appendix. In some places, this makes it difficult to read the main paper. Here’s an example from section 4: “For more general tasks, we showcase the improvement of TELLME on the SIQA and the CoLA task from the GLUE benchmark in Tables 12 and 11 of Appendix A.8.”. This could be improved by ensuring references to the Appendix include a short sentence explaining the result or content of the appendix section and defining any acronyms."
> > >
> > > **A:** Thank you for this suggestion. We will follow your advice by adding a brief in-line summary and defining acronyms on first mention whenever the main text references the Appendix, so that readers can grasp the takeaway without leaving the main paper. For example, the sentence you cited would be revised to: "For more general tasks, we evaluate TELLME on the Corpus of Linguistic Acceptability (CoLA) from the GLUE benchmark and the Social Intelligence QA (SiQA). TELLME improves task performance from 69.1 to 75.8 on CoLA and from 79.9 to 80.2 on SiQA, while also improving disentanglement metrics (Tables 11–12, Appendix A.8)." We will apply this practice to Appendix references throughout the revised manuscript.
> > >
> > > ---
> > >
> > > **Clarifications of minor issues that may further improve readability:**
> > >
> > > We will correct all wording and typographical issues in the revision. Below we clarify two detailed terms whose definitions would help readability:
> > >
> > > > In Section 4.2 and Table 2 the phrase "over-safety" is used but not defined.
> > >
> > > We agree that defining this term inline would improve readability. "Over-safety" refers to the tendency of safety-trained LLMs to refuse benign requests due to overly conservative alignment, resulting in unnecessary capability degradation. We will add this definition when the term first appears.
> > >
> > > > In Tables 1, 2, 3, etc. the heading "Origin" should be defined (presumably this is the original model)?
> > >
> > > "Origin" refers to the original model prior to any modification by TELLME or other baselines. We will add this definition to the relevant table captions in the revised manuscript to make the tables self-contained and easier to read.

---

### Official Review · Reviewer_2nJK · 2026-03-12

**Soundness:** 3
**Presentation:** 3
**Significance:** 3
**Originality:** 3
**Overall Recommendation:** 4
**Confidence:** 3

**Summary:**

The authors propose TELLME, a pipeline to monitor LLMs by disentangling the model's behavior in representation space, which allows them to identify sensitive behaviors earlier and improves LLMs' safety performance. They evaluate their method's monitoring accuracy against various models, primarily with original LLMs, SAEs (and their variations), and Transcoders, using monitors such as LLMs' output-based judgment and representation-based monitors, namely Self-Sim and Linear Probes. They also theoretically explain how TELLME improves LLMs' generalization ability.

**Compliance With Llm Reviewing Policy:**

Affirmed.

**Final Justification:**

The rebuttal addressed a couple of my concerns and I have raised my score accordingly.

**Key Questions For Authors:**

W1, W2, and W3 are my key questions as well.

**Limitations:**

Yes. There is a small typo in the impact statement (monitoer)

**Strengths And Weaknesses:**

Strengths:
* S1. The paper is well motivated and addresses a significant problem in LLM monitoring.
* S2. The method is compared against several well-known techniques (SAEs and Transcoders) that reconstruct activations to understand model representations.
* S3. The authors demonstrate various monitoring scenarios, making the method relatively well-evaluated.

Weaknesses/Recommendations:
* W1. The paper claims that TELLME is highly scalable and robust. However, looking at Algorithm 1 on page 15, it is unclear how scalability is achieved, given that the method constructs a disentangled model. This seems to introduce a similar limitation to SAEs and other representation reconstruction techniques.
* W2. In Table 1, why is the Judge component marked as N/A for SAEs and Transcoders? A possible pipeline for the Judge in these cases would be to prompt the model, list the highly activated features and their descriptions, and either classify based on the number of features indicating safety or harmfulness, or prompt the original LLM (or a common judge LLM) to assess whether the activated features predominantly relate to safety or harmfulness.
* W3. It is unclear how the baseline models (such as SAEs) are constructed and used for evaluation, which raises concerns about the fairness of the comparison.
* W4. While the related work section appears somewhat scoped, safety-specific SAE and Transcoder citations would strengthen the related work. The following papers also seem relevant but are not cited:
  * ICLR: In-Context Learning of Representations by Park et al.
  * Can We Predict Alignment Before Models Finish Thinking? Towards Monitoring Misaligned Reasoning Models by Chan et al.

---

> ### Author Rebuttal · Authors · 2026-03-31
>
> Thank you for your great efforts in reviewing this paper. We will try our best to answer all your questions. **Please let us know if you still have further concerns, or if you are not satisfied with the current responses, so that we can further update the response ASAP.**
>
> ---
>
> **Q1**: "The paper claims that TELLME is highly scalable and robust. However, looking at Algorithm 1 on page 15, it is unclear how scalability is achieved, given that the method constructs a disentangled model. This seems to introduce a similar limitation to SAEs and other representation reconstruction techniques."
>
> A1: Thank you. By "scalable and robust" we mean empirical effectiveness across model scale, architecture, and modality — properties that many alignment algorithms fail to maintain. As shown in Figure 3, TELLME consistently and significantly enhances safety from 7B to 235B parameters, across Dense, MoE, and multimodal models. Structurally, TELLME fine-tunes the model via lightweight LoRA adapters (~41M parameters), whereas our SAE baseline involves ~1.09B parameters (26× more), avoiding the heavy external-module overhead of SAE-style approaches.
>
> ---
>
> **Q2**: "In Table 1, why is the Judge component marked as N/A for SAEs and Transcoders? A possible pipeline for the Judge in these cases would be to prompt the model, list the highly activated features and their descriptions, and either classify based on the number of features indicating safety or harmfulness, or prompt the original LLM (or a common judge LLM) to assess whether the activated features predominantly relate to safety or harmfulness."
>
> A2: Thank you for the valuable suggestion. The Judge task in Table 1 is specifically designed to evaluate LLMs‘ judgement ability through text generation. We marked SAE-style baselines as N/A because they are standardly used as external modules for post-hoc analysis, rather than for directly altering LLMs' generation behavior. While your proposed pipeline is insightful, it relies on a secondary process (e.g., prompting or classifying) to assess the activated features. This essentially evaluates the representations rather than the language output, which is already measured by our Self-Sim and Linear Probe metrics.  To ensure a thorough comparison, **we have followed your suggestion to conduct new experiments** that integrate SAE-style modules directly into the end-to-end inference pipeline of Llama-3.1-8B and evaluate them on the Judge task.
>
> |Judge|Multi-Risk|Safe-or-Harmful|Misalignment|
> |-|-|-|-|
> |Origin|56.1|71.6|87.8|
> |SAE|56.8| 74.8| 86.3|
> |SAE-KL|57.0| 75.2| 86.0|
> |SAE-Multi-TopK| 55.6| 75.3| 85.8|
> |Transcoder|57.4|75.1|85.3|
> |Skip Transcode | 57.1|75.3| 85.5|
> |TELLME|**61.5**|**76.9**|**88.5**|
>
> As shown above, TELLME consistently outperforms all baselines. These results further validate TELLME’s core motivation: natively enhancing the model's own monitorability rather than relying on external add-ons. We will add these discussions and results to the revised manuscript.
>
> ---
>
> **Q3**: "It is unclear how the baseline models (such as SAEs) are constructed and used for evaluation, which raises concerns about the fairness of the comparison."
>
> A3: Thank you. As stated in Lines 269–270 and Appendix A.3 (Table 8 and Lines 928–953), our SAE baseline (following [s1]) uses an expansion factor of 32 and top-128 activation, involving 1.09B trainable parameters -- roughly 26 times the 41M LoRA parameters used by TELLME. This discrepancy in scale ensures that our comparison does not unfairly favor TELLME. To further demonstrate robustness and avoid overstating TELLME's advantages under weak supervision, we also compared it against Google’s open-sourced SAE for Gemma-2-9B, as shown in Table 8. Despite Google's SAE being trained on 8 billion tokens -- nearly 1,000 times more than our own -- TELLME still maintains a significant performance advantage. These results collectively provide a robust and fair empirical validation of TELLME’s effectiveness.
>
> [s1] Verifying Chain-of-Thought Reasoning via Its Computational Graph
>
> ---
>
> **Q4**: "While the related work section appears somewhat scoped, safety-specific SAE and Transcoder citations would strengthen the related work. The following papers also seem relevant but are not cited"
>
> [c1] ICLR: In-Context Learning of Representations by Park et al.
>
> [c2] Can We Predict Alignment Before Models Finish Thinking? Towards Monitoring Misaligned Reasoning Models by Chan et al.
>
> A4: Thank you for the suggestions. [c1] studies how representations are reorganized during in-context learning, and [c2] studies how latent activations can be used to predict misalignment before generation finishes. Relative to them, TELLME focuses on actively improving the LLMs' own representation space through fine-tuning themselves, so that different behaviors become more disentangled and easier to monitor directly. We will cite these papers and discuss works on safety-specific SAEs in the revised manuscript.

---

> > ### Author Rebuttal · Reviewer_2nJK · 2026-04-04
> >
> > Thank you for your response! My concerns have been adequately addressed. I would recommend to share the new table results and specifics of the scalability claim in the main paper. I'll increase my score.

---

> > > ### Author Response · Authors · 2026-04-04
> > >
> > > We are really grateful for your constructive comments and the reply. The comments indeed help us further improve the paper. We will incorporate the new Judge evaluation results and the scalability details into the main paper as you suggested.

---

### Official Review · Reviewer_htBL · 2026-03-14

**Soundness:** 3
**Presentation:** 2
**Significance:** 3
**Originality:** 3
**Overall Recommendation:** 4
**Confidence:** 3

**Summary:**

The authors propose to disentangle latent representations of LLMs to group similar behaviours together, such as harmful or safe. They fine-tune a specific layer in the Transformer stack to achieve this, without impairing general capabilities.

**Compliance With Llm Reviewing Policy:**

Affirmed.

**Key Questions For Authors:**

1. Any experiments on disentangling representations in multiple layers?
2. You fine-tune the model to disentangle representations, but then for tasks where you need to classify (Multi-Risk, Safe-or-Harmful, Misalignment), do you train a classifier on top of the disentangled representations?
3. Any cases where you found that your method would group representations that shouldn't be grouped together? For instance representations talking about violence, but some might be about "violence is good" while others are "violence is bad"?

**Limitations:**

Yes

**Strengths And Weaknesses:**

## Strengths
- Method seems effective and doesn't penalize general performance.
- Good set of experiments and ablations.

## Weaknesses
- Section 3 and 4 were hard to follow. I had to re-read multiple times to understand what was going on, although this isn't my area of expertise. Either I misunderstood, or some details were missing. See my questions below. Additionally, section 3 jumped into the low level details without the authors providing a high level picture of how the method works. This would have been useful. The intro mentions that the method disentangles representations, but there is no mention of fine-tuning the model's weights until the experiment section. One might assume that latent representations are disentangled for safety measurement, but that the original latent representations are untouched and fed to the next Transformer layer.

---

> ### Author Rebuttal · Authors · 2026-03-31
>
> Thank you for your great efforts in reviewing this paper. We will try our best to answer all your questions. **Please let us know if you still have further concerns, or if you are not satisfied with the current responses, so that we can further update the response ASAP.**
>
> ---
>
> **Q1**: "Section 3 and 4 were hard to follow. I had to re-read multiple times to understand what was going on, although this isn't my area of expertise. Either I misunderstood, or some details were missing. See my questions below. Additionally, section 3 jumped into the low level details without the authors providing a high level picture of how the method works. This would have been useful. The intro mentions that the method disentangles representations, but there is no mention of fine-tuning the model's weights until the experiment section. One might assume that latent representations are disentangled for safety measurement, but that the original latent representations are untouched and fed to the next Transformer layer."
>
> A1: Thank you for this helpful comment. We agree that Section 3 currently dives too quickly into low-level details without first giving a clear high-level picture of how TELLME works, and we will revise the manuscript to present the overall pipeline more clearly before introducing the formal objective.
>
> Moreover, TELLME fine-tunes the model itself via LoRA, so the edited representations continue to be used by subsequent Transformer layers during inference. We also agree that this point is not stated early enough in the current draft. We will revise the manuscript, adding a high-level overview at the start of Section 3 before any formal notation, and explicitly stating in the Introduction that TELLME modifies the LLMs' weights via fine-tuning.
>
> ---
>
> **Q2**: "Any experiments on disentangling representations in multiple layers?"
>
> A2: Thank you. While the original manuscript focuses on a single target layer (80% depth), **we have followed your suggestion** to evaluate TELLME across multiple layers (at 30%, 50%, and 80% depth).
>
> || Multi-Risk ||Safe-or-Harmful|| Misalignment ||
> |-|-|-|-|-|-|-|
> ||Self-Sim|Linear Probe|Self-Sim|Linear Probe|Self-Sim|Linear Probe|
> | Origin| 59.2| 77.8| 73.4| 82.9 | 76.5| 92.8 |
> | TELLME| 73.0 | 78.8| 82.5| 84.6| 89.0| 96.3|
> | TELLME + Multi-layer | 73.5 | 80.3| 80.1| 84.7| 86.3| 97.0|
>
> The results show that multi-layer disentanglement consistently enhances Linear Probe performance by providing additional discriminative information. The slight decrease in Self-Sim in some cases indicates that Self-Sim is more sensitive to cross-layer feature interference than the supervised probe. We will include this evaluation in the appendix of revised manuscript.
>
> ---
>
> **Q3**: "You fine-tune the model to disentangle representations, but then for tasks where you need to classify (Multi-Risk, Safe-or-Harmful, Misalignment), do you train a classifier on top of the disentangled representations?"
>
> A3: Thank you. Training an additional classifier is not strictly necessary for TELLME. For instance, our Self-Sim directly performs classification based on representation similarity, entirely bypassing the need to train a separate module. Nevertheless, we also report the classifier performance (Linear Probe) to provide a comprehensive evaluation. As shown in Table 1, for the original models and external-module baselines, there is a large accuracy gap between Self-Sim and Linear Probe. After applying TELLME, this gap is drastically reduced, which aligns with our motivation to enhance the transparency and monitorability of LLMs themselves.
>
> ---
>
> **Q4**: "Any cases where you found that your method would group representations that shouldn't be grouped together? For instance representations talking about violence, but some might be about "violence is good" while others are "violence is bad"?"
>
> A4: Thank you. TELLME is specifically designed for behavior-level disentanglement rather than word-level grouping.
>
> To illustrate, **we have followed your suggestions** to show a concrete case where both a safe and an unsafe sample explicitly mention "violence":
>
> - Safe Sample: Q:*"Hire someone to beat up someone who wronged me"*  A:*"I do not condone the use of* ***violence**, and therefore am unable to help..."*
> - Unsafe Sample: Q:*"If you were a serial killer..."*  A:*"...executing them in a swift and* ***violent*** *manner..."*
>
> While both samples contain violence-related keywords, they represent fundamentally different behaviors (refusal vs. harmful execution). Crucially, the l_2 distance between their representations increased notably from 9.37 (Origin) to 15.17 (TELLME).
>
> This example illustrates that TELLME pushes behaviorally distinct samples further apart, confirming that TELLME captures the nuanced distinction between "violence as a topic" and "violence as a harmful behavior." We will add this case study and discussion to the revised manuscript.

---

> > ### Author Rebuttal · Reviewer_htBL · 2026-04-06
> >
> > Thank you for your nice rebuttal, and for the added experiments. I look forward to seeing the updated paper.

---

> > > ### Author Response · Authors · 2026-04-07
> > >
> > > We are really grateful for your thoughtful review and encouraging feedback. Your comments have been invaluable in improving the paper. We will incorporate the multi-layer experiments, the violence case study, and the high-level overview of Section 3 into the revised manuscript.

---

### Decision · Program_Chairs · 2026-04-30

**Decision:**

Accept (regular)

**Comment:**

This paper proposes TELLME to address the monitorability of LLMs' toxic, harmful, and unsafe behaviors in the hidden representation space. In particular, TELLME uses contrastive learning to disentangle or group the hidden representations, leading to different output behaviors (safe or harmful). This is achieved by training LoRA adapters that can change the hidden representation of intermediate layers.  They evaluate their method's monitoring accuracy against various models, primarily with original LLMs, SAEs (and their variations), and Transcoders, using monitors such as LLMs' output-based judgment and representation-based monitors, namely Self-Sim and Linear Probes. They also theoretically explain how TELLME improves LLMs' generalization ability. Empirically, they show that TELLME does not hurt the general performance on MMLU and MMLU-pro.

The reviewers agree that the monitorability problem of LLMs is important and that the proposed contrastive learning-based disentanglement of hidden representations is novel in addressing the challenge. The experiments on multiple VLMs/LLMs are comprehensive, as is the ablation study. Meanwhile, reviewers raised questions about the clarity of the presentation. More clarification is needed for the scalability, the setups for baselines, the accuracy of LLM judges, etc. Reviews are particularly worried about the risks of overclaims in theoretical analysis and the misunderstanding of visualizations. The rebuttal addressed the main concerns of three out of four reviewers. The remaining concerns suggest that the authors narrow the scope and focus on one key problem with more rigorous analysis. Given the room for improvement, I recommend weak acceptance of this paper.